# Structure and dynamics of the *E. coli* chemotaxis core signaling complex by cryo-electron tomography and molecular simulations

C. Keith Cassidy [1,2]*, Benjamin A. Himes[3], Dapeng Sun [3], Jun Ma[3], Gongpu Zhao [3], John S. Parkinson [4], Phillip J. Stansfeld [1,5], Zaida Luthey-Schulten [2,6] & Peijun Zhang [3,7,8]*

To enable the processing of chemical gradients, chemotactic bacteria possess large arrays of transmembrane chemoreceptors, the histidine kinase CheA, and the adaptor protein CheW, organized as coupled core-signaling units (CSU). Despite decades of study, important questions surrounding the molecular mechanisms of sensory signal transduction remain unresolved, owing especially to the lack of a high-resolution CSU structure. Here, we use cryo-electron tomography and sub-tomogram averaging to determine a structure of the *Escherichia coli* CSU at sub-nanometer resolution. Based on our experimental data, we use molecular simulations to construct an atomistic model of the CSU, enabling a detailed characterization of CheA conformational dynamics in its native structural context. We identify multiple, distinct conformations of the critical P4 domain as well as asymmetries in the localization of the P3 bundle, offering several novel insights into the CheA signaling mechanism.

[1] Department of Biochemistry, University of Oxford, Oxford OX1 3QU, UK. [2] Department of Physics and Beckman Institute, University of Illinois at Urbana-Champaign, Urbana, IL 61801, USA. [3] Department of Structural Biology, University of Pittsburgh School of Medicine, Pittsburgh, PA 15260, USA. [4] School of Biological Sciences, University of Utah, Salt Lake City, UT 84112, USA. [5] School of Life Sciences & Department of Chemistry, University of Warwick, Coventry CV4 7AL, UK. [6] Department of Chemistry and Center for the Physics of Living Cells, University of Illinois at Urbana-Champaign, Urbana, IL 61801, USA. [7] Division of Structural Biology, Wellcome Trust Centre for Human Genetics, University of Oxford, Oxford OX3 7BN, UK. [8] Electron Bio-Imaging Centre, Diamond Light Sources, Harwell Science and Innovation Campus, Didcot OX11 0DE, UK. *email: keith.cassidy@bioch.ox.ac.uk; peijun@strubi.ox.ac.uk

Motile bacteria are able to seek out optimal physiological conditions through a behavior known as chemotaxis, where highly conserved signal transduction pathways couple environmental chemical gradients to cellular motility[1,2]. The streamlined chemotaxis pathway of *Escherichia coli* is especially well studied and provides a powerful tool for investigating the molecular underpinnings of sensory signal transduction[3–5].

Input to the *E. coli* chemotaxis pathway is mediated by transmembrane chemoreceptors, which operate as homodimers and respond to specific sets of external chemoeffectors[4,6]. Upon ligand occupancy changes, chemoreceptors transmit signals across the inner membrane that modulate the autophosphorylation activity of a homodimeric histidine kinase CheA, which is coupled to receptor control by an adaptor protein CheW. In addition to ligand binding, the methylation state of four cytoplasmic glutamyl or glutaminyl residues, denoted QEQE in wild-type receptors, adjusts output activity[6]. Addition or removal of methyl groups at these sites via specific enzymes, modulates the overall level of CheA activity to offset the effects of ligand binding inputs. Importantly, Q residues mimic the effects of methylated E residues on CheA activity, so a receptor with all four modification sites as glutamine (4Q) promotes high CheA activity, whereas a 4E receptor produces low kinase activity[7–9].

Each CheA monomer comprises five domains, P1–P5. The P3, P4, and P5 domains are connected by two short linkers[10] and enable dimerization, ATP hydrolysis, and binding to chemoreceptors and CheW, respectively. The P1 and P2 domains, which are connected to the P3–P4–P5 dimer core by long flexible linkers, mediate the transfer of phosphoryl groups from P4 to P1 and from there to response regulators[11,12]. The CheY response regulator binds to the flagellar motors upon phosphorylation to affect their rotational bias and ultimately cellular swimming pattern.

CheA regulation requires the assembly of core-signaling units (CSUs)[4,13,14] with a well-defined structure, comprising two chemoreceptor trimer-of-dimers (TOD), a single CheA dimer and two CheW monomers (Fig. 1d). Upon formation of the CSU, CheA undergoes pronounced tertiary rearrangement[15–17], increasing its basal activity nearly three orders of magnitude[12,18]. CSUs further cluster into large, hexagonally packed chemosensory arrays, which place each CheA under the influence of as many as 20 receptors, giving rise to highly cooperative signaling responses[15,19–21]. The basic organizational hierarchy of the CSU and the extended hexagonal architecture it forms appear to be universally conserved features of bacterial and archaeal chemotaxis pathways[22,23]. Considering the central role that chemotaxis plays in the infection process of numerous human and plant pathogens[24–27], a detailed characterization of CSU structure and function would, therefore, provide an excellent source of novel targets for the development of broadly applicable antimicrobial therapies.

Despite a considerable effort, the large size and dynamic nature of the CSU have thwarted the determination of a high-resolution structure via conventional structural biology techniques such as X-ray crystallography, nuclear magnetic resonance, and single-particle cryo-electron microscopy. Previous studies involving cellular cryo-electron tomography (cryoET) have provided snapshots of native chemosensory arrays in varying signaling states, considerably advancing our understanding of chemosensory array architecture, assembly, and function[15,19,22,28,29]. However, cell thickness, even that of minicells, typically limits the resolution of in vivo structures to 20 Å or lower[30]. Hence, detailed descriptions of the molecular mechanisms underlying sensory signal transduction, especially that of receptor-mediated CheA control, remain elusive.

Here, we present in vitro cryoET structures of the QEQE, 4Q, and 4E *E. coli* CSU, derived using our recently developed emClarity software[31] package for high-resolution cryoET and sub-tomogram averaging (STA). For the 4Q system, we achieve an overall sub-nanometer resolution, which although anisotropic, enables the construction of an all-atom model of the *E. coli* CSU. Using molecular simulations, we further investigate the conformational landscape of CheA within the CSU, identifying multiple, distinct conformations of the critical CheA.P4 domain as well as asymmetric conformations of the CheA.P3 bundle, highlighting stabilizing features of each. The implications of our results for the CheA signaling mechanism are discussed.

## Results

### CryoET structures of in vitro reconstituted chemosensory arrays.

We previously developed an in vitro reconstitution system comprising *E. coli* CheA.P3–P4–P5, CheW, and a His-tagged cytoplasmic domain of wild-type Tar (TarCF) bound onto Ni$^{2+}$-NTA lipid containing monolayers (Fig. 1e)[16,32,33]. This system produces samples that are suitable for the high-resolution structural analysis of chemosensory arrays by cryoET. In particular, compared to native cellular arrays, in vitro monolayer arrays are thin (~25 nm), compositionally well defined, and homogeneous. Moreover, the abundance of in vitro arrays allows one to extract a few thousand CSUs from a single tomogram, enabling the use of STA to obtain high-resolution structures of the CSU. We have now reconstituted in vitro monolayer systems using 4Q or 4E variants of the aspartate receptor signaling domain (TarCF), giving rise to TarCF$_{4Q}$-CheA.P3.P4.P5-CheW and TarCF$_{4E}$-CheA.P3.P4.P5-CheW monolayer arrays, which should display high and low kinase activity, respectively.

As depicted in Fig. 1a–c, 4Q arrays appear better ordered compared to QEQE arrays (i.e., show larger patches of extended, well-packed CSUs), and both appear highly ordered compared to 4E arrays. The decreased long-range order observed in 4E arrays is generally consistent with previous studies that show the CheA-OFF array state exhibits reduced cooperativity[8,34,35], likely through the disruption of interactions between CSUs that alter the tightness of their physical and functional coupling[36]. While in vivo cryoET has previously shown that arrays remain intact and hexagonally packed during signaling[37], the detailed long-range order of CSU packing has not been examined at the tomogram level, making is difficult to say whether in vivo arrays might not also exhibit variations in the size of well-packed patches. At present, however, we cannot rule out the possibility that the differences in order observed in the monolayer array may arise or appear more pronounced due to their in vitro nature. Therefore, the signaling role played by such changes, if any, will require additional study.

We collected 24, 22, and 19 cryo-tomography tilt series of the 4Q, QEQE, and 4E monolayer arrays, respectively (Supplementary Table 1). CSUs were identified within each tomogram using template matching as implemented in emClarity (Supplementary Fig. 1). The resulting CSU sub-tomograms were extracted and processed with emClarity using an iterative approach to refine both the tilt geometry and alignment parameters of each sub-tomogram[31]. Further classification in emClarity was performed to differentiate CSU densities from the neighboring dimers of receptor TODs that lacked CheA density. The three-dimensional (3D) density maps of the 4Q, QEQE, and 4E CSUs derived from STA are shown in Fig. 1 (f–h); the unbiased Fourier shell correlations (FSCs) indicate the overall resolutions of these density maps are 8.4 Å, 10.1 Å, and 14.5 Å, respectively (Supplementary Fig. 2a). Clearly, the relatively well-ordered 4Q and QEQE arrays have produced considerably better-resolved density maps than the disordered 4E arrays.

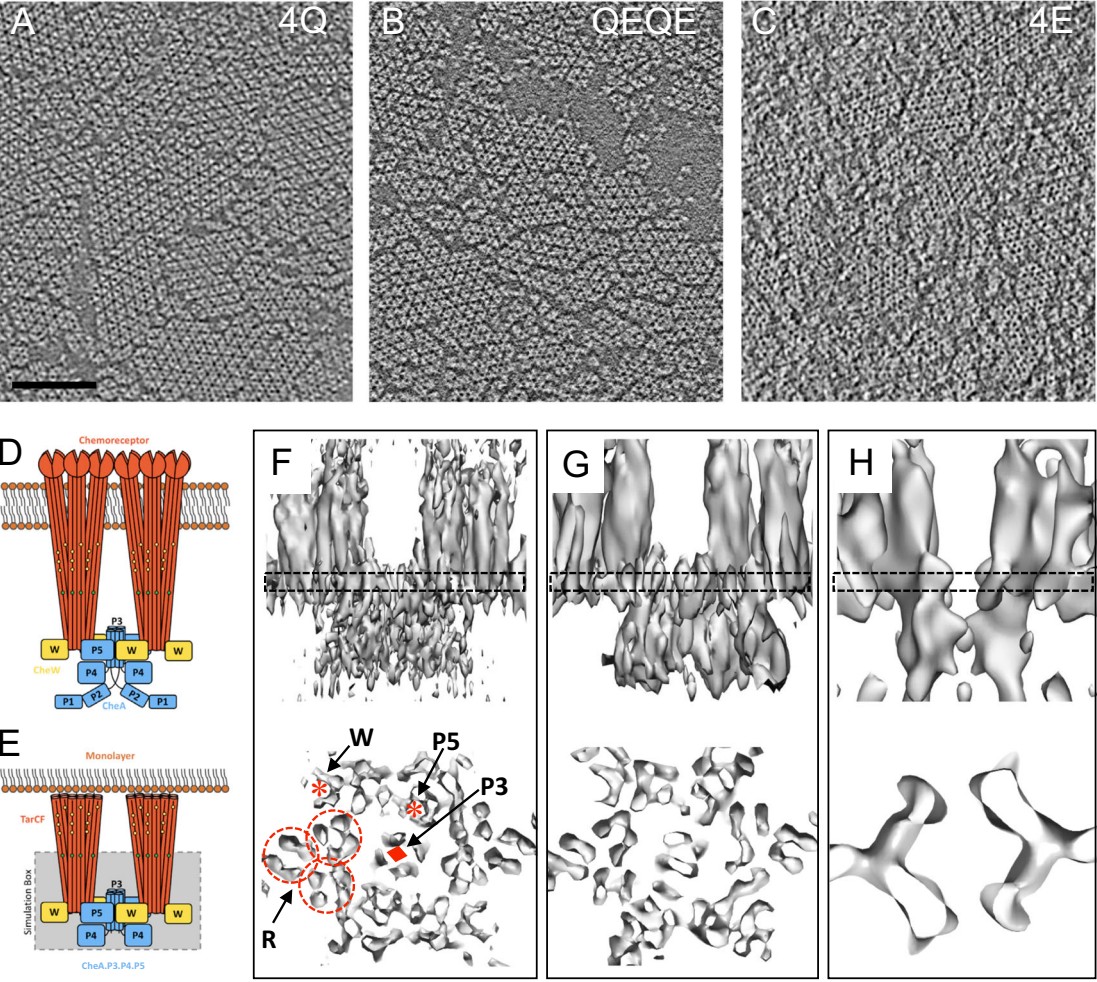

**Fig. 1 Cryo-electron tomography and sub-tomogram averaging of in vitro reconstituted chemosensory arrays. a–c** Raw tomographic slices of monolayer arrays of TarCF$_{4Q}$/CheA/CheW **a**, TarCF$_{QEQE}$/CheA/CheW **b**, and TarCF$_{4E}$/CheA/CheW **c**. Scale bar, 100 nm. **d, e** Schematics of the transmembrane **d** and in vitro monolayer **e** core-signaling unit (CSU), consisting of two receptor trimer-of-dimers in red, a single CheA dimer in blue, and four CheW monomers—two essential (bound to CheA.P5) and two ancillary (flanking)—in gold. In **e**, a grey box denotes the region of the CSU modeled and simulated in this study. Receptor modification sites, the glycine hinge, and membrane headgroups are depicted as yellow, teal, and orange circles, respectively. **f–h** Sub-tomogram averages of CSU of TarCF$_{4Q}$/CheA/CheW **f**, TarCF$_{QEQE}$/CheA/CheW **g**, and TarCF$_{4E}$/CheA/CheW **h**. 3D volumes are displayed in surface rendering as side views (top) and top views (bottom). Top views are clipped from the boxed regions surrounding the CheA/CheW baseplate region. In **f**, red asterisks indicate the CheA.P5 and CheW beta barrels, a red diamond indicates the CheA.P3 four-helix bundle, and dashed circles indicate the individual receptor dimers within a trimer.

At the current resolution, the 4Q and QEQE CSU maps show marginal differences overall (Supplementary Movie 1). The high resolution of the 4Q density map, however, reveals several secondary structure features. These include individual alpha helices in four-helix bundles of the chemoreceptor dimers (Fig. 1f, dashed circles) and the CheA.P3 dimerization domain (Fig. 1f, diamond), as well as beta barrels in both the CheA.P5 domain and CheW (Fig. 1f, asterisks). Moreover, as observed previously in the QEQE monolayer system[16], all three receptors within each TOD, have a close interaction with either CheA.P5 or CheW in a ratio of 1TOD:1CheA:2CheW, confirming the existence of an additional CheW monomer flanking each TOD (Supplementary Fig. 3d, Fig. 1d). In the context of the extended array, these additional CheW monomers form rings that act to network neighboring CSUs[16,19], thus, reinforcing the notion that CheW plays an important functional role outside the CSU, namely through the formation of complete or partial CheW-only rings that contribute to the ultra-stability and high cooperativity of the chemosensory array[38,39]. The least resolved portion of the 4Q map corresponds to the CheA.P4 domains, indicating

substantial conformational heterogeneity in this region of the CSU.

**Atomistic model of the *E. coli* CSU.** To enable a detailed investigation of CSU structure and dynamics, we constructed an atomistic model of the *E. coli* complex, using the 8.4 Å-resolution 4Q density map to explicitly guide the modeling process[40]. Owing to their lower resolution, the QEQE and 4E maps did not provide additional structural information and were not used for subsequent modeling. Component models of the receptor TOD, CheA.P3.P4 dimer, and CheA.P5-CheW (interface 1) dimer were first generated to make the best use of existing structural information at protein–protein interfaces (Fig. 2a). As no high-resolution structures exist of the cytoplasmic portion of *E. coli* Tar, we instead used a crystal structure of the *E. coli* Tsr TOD (PDB 1QU7[41]), which shares a 96% sequence similarity with Tar. Models of the *E. coli* CheA.P3.P4 dimer and the CheA.P5-CheW dimer were generated using template-based homology modeling from crystal structures of *Thermotoga maritima* CheA and CheW

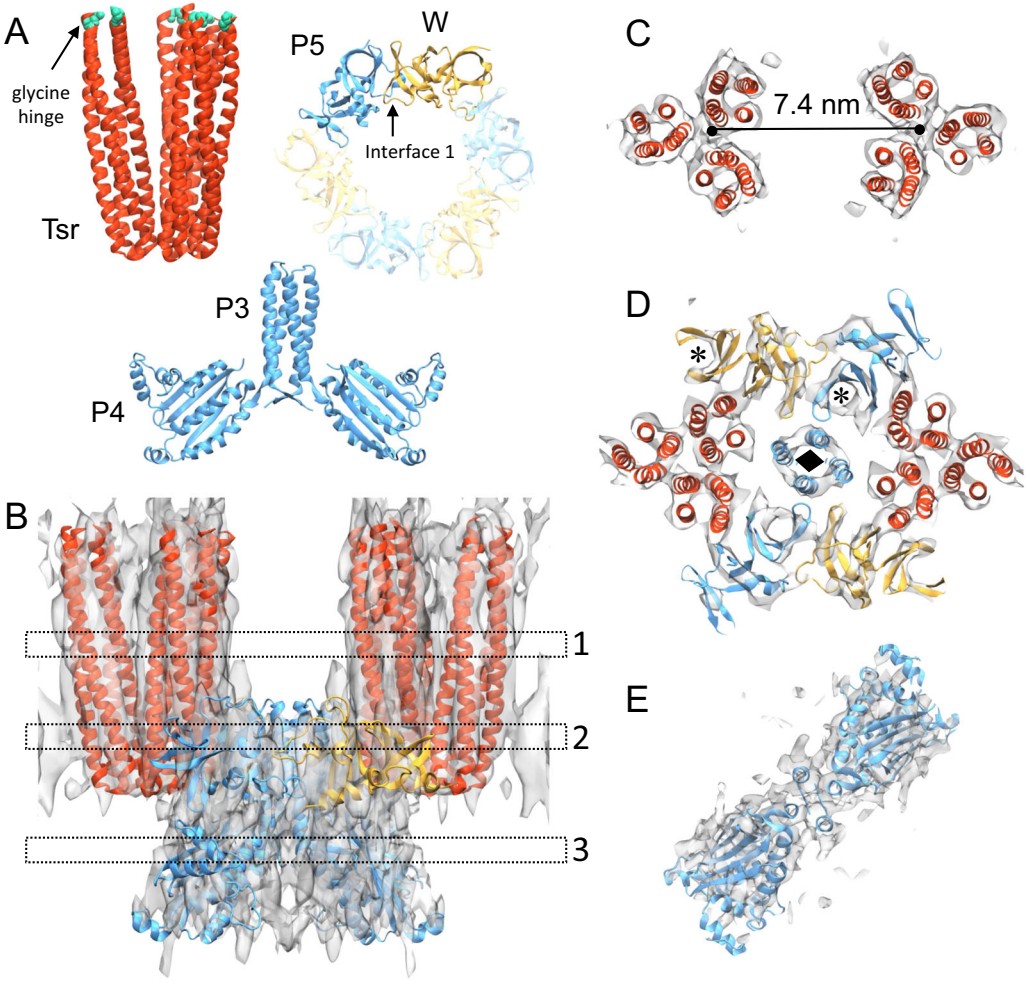

**Fig. 2 All-atom model of the *E. coli* CSU. a** Component models used for initial CSU assembly, including the protein interaction region of the Tsr receptor trimer-of-dimers (TOD) and homology models of the CheA.P3-P4 and CheA.P5-CheW interface 1 dimers. Receptors are colored in red, CheA in blue, and CheW in gold. The residues comprising the glycine hinge are shown in teal. **b** Overlay between the 4Q density map, shown in surface rendering with transparent grey, and the MDFF-refined CSU model. **c–e** Sectional views of the boxed regions in **b** (1–3), highlighting the quality-of-fit at the level of the receptor TODs (1), CheA–CheW baseplate (2), and CheA.P4 domains (3). The distance between the symmetry axes of the receptor TODs is denoted in **c**. In **d**, black asterisks indicate the CheA.P5 and CheW beta barrels; a black diamond indicates the CheA.P3 four-helix bundle.

(PDBs 4XIV[17] and 4JPB[42], respectively), which are sufficiently similar (75% sequence similarity for CheA.P3.P4.P5; 55% for CheW) to permit the construction of reliable models (Methods section). Segmentation of the 4Q map into sub-densities corresponding to each component then provided a basis for their initial rigid docking.

The best-resolved areas of density belong to the receptor TODs, where each receptor can be clearly differentiated, and individual helices are discernable up to the glycine hinge (Fig. 2b). We began assembling the CSU by optimizing the position and conformation of the protein interaction region (residues 340–441) of the receptor TODs using Molecular Dynamics Flexible Fitting (MDFF)[43,44]. The resulting fits minimally altered the conformation of the initial models (backbone root-mean-squared-deviation (RMSD) = 1.8 Å) and oriented the two TODs symmetrically with a separation of 7.4 nm (Fig. 2c). This spacing suggests an extended hexagonal lattice constant of ~12.8 nm, between the 12 nm[22] and 13.2 nm[19] spacings previously reported. To test the robustness of the obtained fit, especially considering resolution anisotropy (Supplementary Fig. 2b), we generated different TOD starting positions by translations of up to ±15 Å along the Z-direction and refitted the models. In all cases, the

TOD structure converged to the same position within the density (Supplementary Fig. 4, Supplementary Movie 2).

The densities corresponding to CheA.P5 and CheW were also well resolved, showing distinct beta barrel organizations coinciding with subdomain 1 of each protein. We docked CheA.P5 and CheW independently without constraints on interface 1. The resulting positioning produced a nearly identical configuration to that seen in crystal structures[15,42] (backbone RMSD = 1.4 Å), suggesting this interface 1 binding mode is prominent in the 4Q map. Finally, docking the CheA.P3.P4 dimer model produced a nonoverlapping structure while also situating the P4 domain such that it could be readily connected to the P5 domain via the short P4–P5 linker. The entire complex was then subjected to MDFF, producing minor refinements to the positioning of the CheA.P5-CheW and CheA.P3.P4 models. The resulting CSU model is shown in Fig. 2(b–e) and Supplementary Movie 3.

**CheA.P4 adopts multiple, distinct conformations within the CSU.** A growing body of evidence suggests that CheA is highly dynamic, undergoing pronounced interdomain rearrangements during CSU assembly and throughout its catalytic cycle[11,17,45,46].

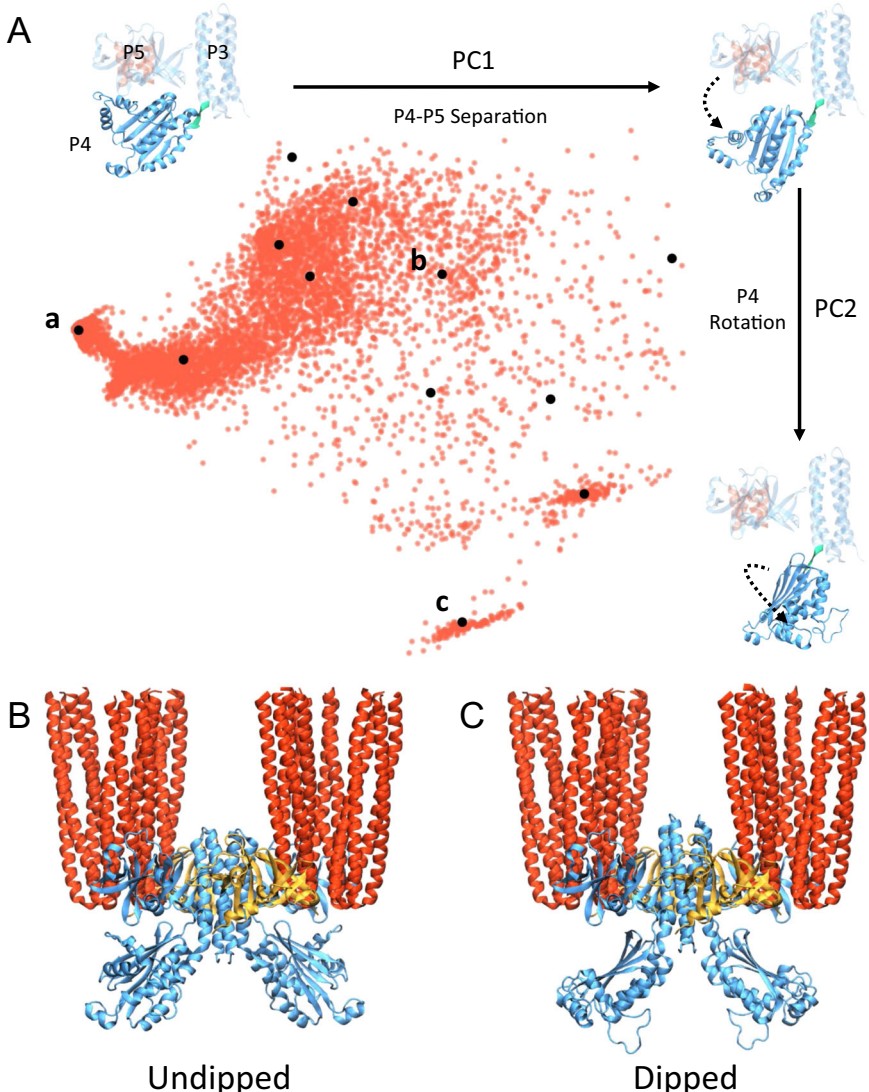

**Fig. 3 CheA.P4 adopts multiple distinct conformations within the CSU. a** Projection of the 9247 GSA-derived P4 conformations onto the first two PCs of the ensemble (red dots). Black dots denote the 12 medoid structures identified by clustering analysis of the ensemble. Three medoid projections and corresponding structures are shown, illustrating the primary degree of freedom represented by each PC. In each structure, the portion held static during the GSA simulations is shown with a transparent representation and the P3–P4 hinge (residues 326–328) about which rigid-body rotations were made is shown in teal. **b–c** CSU models containing a representative MDFF-refined undipped **b** and dipped **c** CheA dimer.

Indeed, within the 4Q map, the density corresponding to the CheA.P4 domain is considerably less resolved, suggesting substantial conformational heterogeneity. We previously used molecular dynamics (MD) simulations to identify an alternative P4 conformation in *T. maritima* CheA, which we termed dipped and subsequently validated in live *E. coli* cells[16]. Thus, to systematically explore the conformational landscape of *E. coli* CheA in its native structural context, we used generalized simulated annealing (GSA)[47,48] to generate an ensemble of P4 positions within our CSU model through rigid-body rotations about the P3–P4 hinge (residues 326–328). The full details of our GSA protocol are provided in the Methods section.

We analyzed a total of 9247 GSA-derived structures. Figure 3a shows each structure projected onto the first two principal components (PCs) of the full ensemble, which capture 89% of its total variance. The resulting distribution clearly illustrates that as P4 becomes increasingly separated from P5 (movement along PC1), it is able to adopt a much wider range of rotational states (movement along PC2) in which the active site and ATP-lid are

markedly reoriented with respect to the rest of the CSU. This suggests that the regulatory ability of P5 may arise from its capacity to constrain P4 to a highly specific orientation through direct contact, which if disrupted could allow P4 to access a much broader conformational landscape. Indeed, construction of CheA dimer models based on the GSA ensemble shows that P4–P4 separation, as measured by the distance between active sites, can vary substantially (~35–70 Å; Supplementary Fig. 5). In particular, at distances of less than ~40 Å, the P4 domains come close enough to directly interact (Supplementary Fig. 5d), a possibility previously reported[11,45].

We then considered whether the P4 density in our 4Q map could account for additional P4 conformations. To define a subset of the GSA-derived conformations that preserved the structural variability of the full ensemble, hierarchical clustering was used to group conformations based on pairwise RMSD[49]. This analysis resulted in 12 structures with a wide range of P4 positions (Supplementary Fig. 6). Symmetric CheA dimers constructed from each were then substituted into our CSU model and MDFF

was used to refit the P4 domains to the 4Q map. Strikingly, the resulting fits collapsed into just two classes of CheA conformation with similar P4 CCC (Supplementary Fig. 7a). We term these classes undipped and dipped to denote the relative position of P4 with respect to the rest of the CSU structure (Fig. 3b, c). A third, intermediate class was also observed (Supplementary Fig. 7b–d), which is qualitatively similar to the undipped class. We note that the symmetry of individual P4 monomers within the obtained conformational classes stems from the symmetric nature of the CSU density map, a product of the STA procedure, and does not imply that the P4 domains must move in unison.

The undipped structures essentially reproduce the initial MDFF-refined CheA conformation, showing a relatively planar P3–P4 configuration with considerable P4–P5 interaction. The dipped structures, on the other hand, do not exhibit a planar P3–P4 conformation, but rather show the P4 domains completely separated from P5 and rotated such that both active sites face downward and are separated by ~55 Å. These structures closely resemble the dipped conformation previously predicted in *T. maritima* CheA[16], providing further evidence that it is a highly conserved, and likely integral, conformational feature of the CheA catalytic cycle. Considering the appreciable difference in orientation of the P4 active sites in the planar, undipped, and intermediate conformations compared to the rotated, dipped conformation, it's likely that transitions between these conformations play a role in regulating P1/P4 interactions necessary for phosphotransfer. Interestingly, Greenswag et al. have suggested that the inhibited state of CheA may involve close association between both P1 and P4 domains in a dimer[17]. Thus, the decreased separation between dipped P4 domains may provide a substrate for formation of inhibiting P1–P4 interactions across the dimer, which are not possible between undipped P4 conformations.

**Structural elements stabilizing alternate CheA.P4 conformations**. To assess the stability of the identified CheA classes and probe the dynamics of the CSU, we conducted $3 \times 250$-ns all-atom MD simulations of a representative CSU structure from the undipped and dipped CheA classes. The most striking differences are observed at the P3–P4 interface where an antiparallel beta sheet interaction involving residues M327 and L362 in the undipped state (Fig. 4a–b) was disrupted by GSA in the dipped state (Fig. 4c). This permits the formation of a continuous, albeit kinked, helical connection between the two domains during equilibration of the dipped CSU model (Fig. 4c), which appears to stabilize the dipped P4 conformation and is accompanied by a shift in salt-bridge patterns, involving several highly conserved residues (Supplementary Table 2, Supplementary Fig. 8a). In particular, in the undipped simulations, R325 forms salt bridges with D363 and D272, while K270 forms a weaker contact with E361 in several monomers (Fig. 4d); these interactions are in line with those previously seen in crystal structures of *T. maritima* CheA[10,17], including a catalytically active P3–P4 dimer[17]. In the dipped simulations, however, the rotational orientation of the P4 domain disfavors the R325–D363 and K270–E361 interactions, leading instead to the formation of a salt bridge between R325 and E361 as well as increased interaction between K364 and D272 (Fig. 4e, Supplementary Table 2). Notably, these changes involve or directly affect the P3–P4 linker (residues 322–327) that was previously shown to be crucial for CheA activation and basal activity[46,50], further suggesting that transitions between undipped and dipped P4 conformations play a critical role in the CheA catalytic cycle.

In addition, as noted previously, the degree of interaction between the P4 and P5 domains differs considerably between the undipped and dipped conformations. In particular, whereas P4 makes few contacts with P5 in the dipped simulations, interactions between two residue clusters with complementary electrostatics are observed in the undipped simulations (Fig. 4f, Supplementary Table 2). However, although interactions between one or both clusters are often present, the specific interaction partners within a cluster vary. Considering this lack of specificity, as well as the imperfect conservation of certain cluster residues (Supplementary Fig. 8a), it's unlikely that any single P4–P5 interaction cited is critical for CheA function. Rather, both clusters of interactions appear to contribute to the general stabilization of the same planar P4–P5 binding mode, lending further support to the notion that P5 acts principally to constrain P4 orientation.

**CheA.P3 can adopt an asymmetric position within the CSU**. The simulation data also report on the dynamic behavior of the P3 domain. Despite the structural signatures described above, CheA does not adopt strictly symmetric conformations in simulations of either conformational class. In particular, the P3 four-helix bundle is observed to drift and/or lean toward one or the other receptor TOD in multiple undipped and dipped simulations (Supplementary Fig. 9). While these shifts are slight in terms of the center-of-mass movement of the P3 bundle, several asymmetric contacts with the receptors and P5 form repeatedly (Fig. 4g, Supplementary Table 2). In addition, the tight structural coupling between the P3 and P4 domains, both in the undipped and dipped conformations, cause asymmetries in the position of the P3 bundle to be transmitted directly to both P4 domains, giving rise to differing P4–P5 interactions within each monomer (Supplementary Movie 4). It is, thus, conceivable that changes in the position or orientation of the P3 bundle could disengage the P4 domains from other parts of the CSU, in particular P5. As a result, subtle signaling-related rearrangements in CSU structure could alter the aforementioned contacts and/or the effective space between TODs to regulate the magnitude of fluctuations in the position of the P3 bundle and, therefore, the interconversion between key conformations in the CheA catalytic cycle. We note that the receptors were positionally restrained in our simulations to prevent structural artifacts due to their truncation, thus, additional simulations will be required to work out the full details of the effects of receptor conformation on the observed changes in CheA.

**Receptors form strong interactions with CheW but not CheA. P5**. Characterization of the remaining protein–protein interfaces highlighted three strong contacts at the CheW–receptor interface, which were present in the outer submits of the dimers in all simulations: R40–E402, D107–R404, and E18–R394 (Supplementary Fig. 10a, Supplementary Fig. 8b-c). Notably, receptor residues E402 and R404 were previously shown to be critical for stabilizing the CheA-ON state, an effect attributed primarily to a predicted E402–R404 inter-receptor salt bridge[51], which our simulations confirm (Supplementary Fig. 10a). In contrast, the CheA.P5–receptor interface, which is rotated slightly compared to the CheW–receptor interface despite being pseudosymmetric (Supplementary Fig. 10b), does not appear to contain any strong or specific interactions between the two proteins. This is in line with previous studies that suggest the CheA.P5–receptor interface is weaker and only plays a passive role in receptor–CheA communication[52]. Interestingly, receptor residue R404 was shown to play a critical, yet unresolved, role in achieving high cooperativity in the extended array[51]. The D107–R404 salt bridge, which is located directly adjacent to interface 2, suggests an allosteric pathway also involving CheW residue R140 (Supplementary

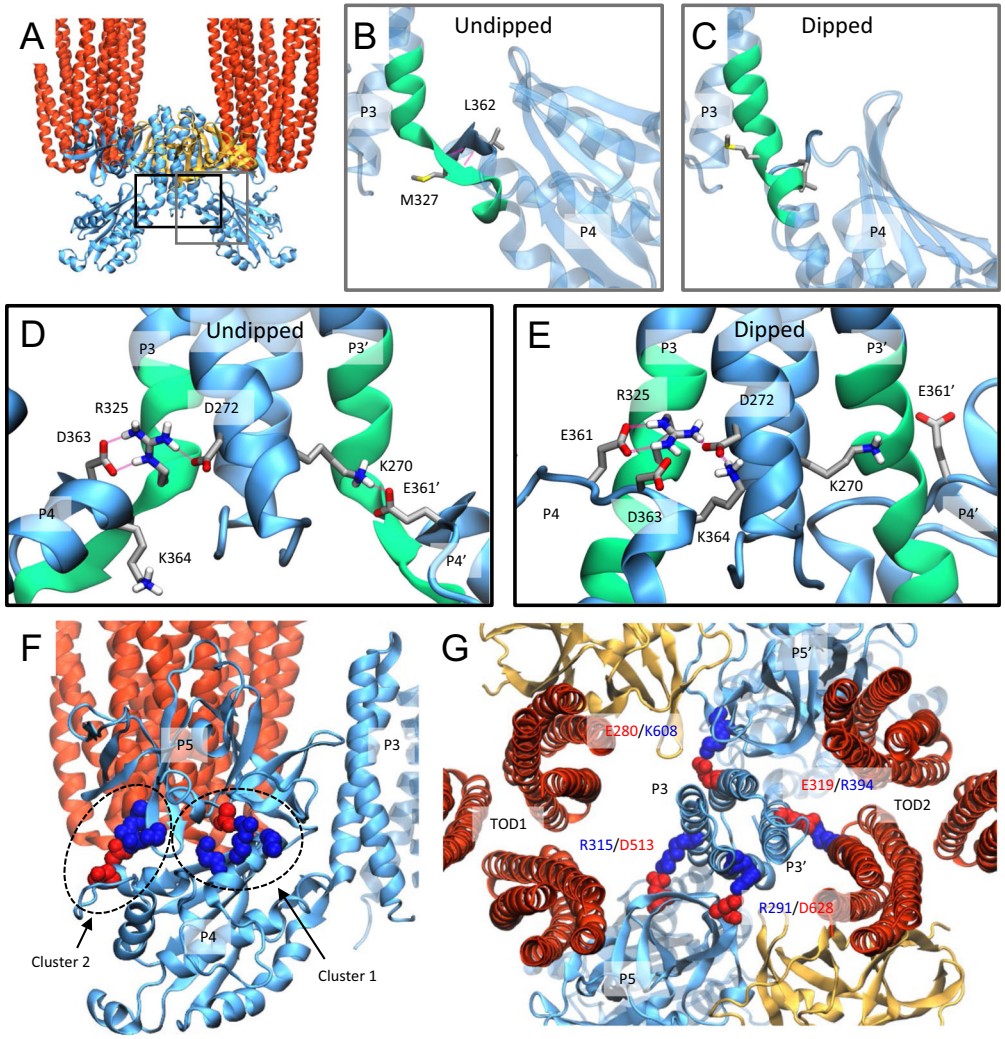

**Fig. 4 Characterization of the undipped and dipped CheA conformational classes using all-atom MD simulation. a–e** Structural changes at the CheA P3–P4 interface stabilizing alternate P4 conformations. **a** Gray and black rectangles denote the location of zoomed **b** and **c** (gray), and **d** and **e** (black) within the context of the larger core-signaling unit (CSU), shown here with an undipped CheA dimer. **b–c** Gray rectangles highlight an observed change in the secondary structure of the P3–P4 connector region (residues 320–332, shown in green), which forms a beta sheet interaction with nearby P4 residues in the undipped conformation **b** and a continuous helix in the dipped conformation **c**. **d–e** Alterations in salt-bridge patterns at the P3–P4 hinge, involving several highly conserved residues (Supplementary Fig. 8), further stabilize the structural changes in the P3–P4 connector region. In **b–e**, only the P3 and P4 domains are shown for clarity. **f** Two clusters of residues with complementary electrostatics stabilize a planar P4–P5 binding mode (Supplementary Table 2). **g** A representative asymmetric P3 conformation, highlighting P3–P5 and P3–receptor interactions differing between CheA monomers.

Fig. 10c), by which R404 exerts influence on interactions between neighboring CSUs. Although the focus in the present manuscript has been on the structure and dynamics of a single CSU, our model may be used to construct higher-order assemblies, enabling a detailed investigation of the conformational coupling between CSUs and the origins of high cooperativity in extended arrays.

## Conclusion

The highly conserved bacterial chemotaxis pathway represents one of the best-studied signal transduction systems in all of biology. Consequently, a wealth of genetic, biochemical, and biophysical data has been assembled, especially in the model organism *E. coli*, which provide excellent opportunities for developing and comparing mechanistic signaling models. Owing to the complexity of the chemosensing machinery, however, high-resolution structural data have proven relatively hard to come by,

hindering integration of the aforementioned insights into a comprehensive signal transduction model. In this work, we present an interdisciplinary approach, where we combine in vitro cryoET and STA with molecular modeling and all-atom simulations to provide detailed, residue-level insight into the dynamics of the CSU.

Our results shed new light on several conformational features of CheA in its native structural context. In particular, we show that the CheA.P4 domain is structurally able to adopt a wide range of conformations within the CSU. Consistent with this observation, the P4 region in the 4Q map is less resolved, suggesting that the P4 domain is, indeed, quite mobile. Nevertheless, flexible fitting simulations revealed two, qualitatively different P4 states, the undipped and dipped conformations. Thus, our data support a signaling model in which CheA regulation involves the selection and stabilization of a few specific conformations from a broad landscape of possible conformations. Moreover, the existence of multiple, distinct conformations within our 4Q

(CheA-ON) cryoET data, additionally, suggests that chemoreceptors do not modulate CheA activity by exclusively isolating one or the other of these critical CheA conformations, but rather by biasing the relative occupancies of each state within the population.

We, additionally, provide key atomistic insights into the structural elements responsible for stabilization of the identified states. In particular, we offer a structural explanation for the role of the P3–P4 linker in CheA signaling, showing that this region is involved in different and specific secondary structural motifs at the P3–P4 hinge, which must interconvert in order to stabilize alternate P4 conformations. Moreover, we predict a number of residue pairs that are unique to either conformation, providing specific targets for mutagenesis and cross-linking experiments as well as potential in vivo reporters, which should enable elucidation of their roles in the catalytic cycle. Finally, we suggest a role for the P3 domain in CheA signaling, namely that through shifts between symmetric and asymmetric positions within the CSU, the P3 four-helix bundle can alter the interactions and localizations of both P4 domains simultaneously to affect the conformational dynamics of the CheA dimer as a whole. In general, the all-atom molecular models emerging from the present study should provide a valuable structural platform on which to compare, integrate, and expand current models of bacterial signal transduction using experimental and computational techniques alike.

## Methods

**Materials**. Plasmids and cell strains used in this study came from the Parkinson lab, except for plasmid pHTCF, which was a kind.pngt from Dr. Weis, University of Massachusetts, Amherst. Plasmid pHTCF is an isopropyl β-d-1-thiogalactopyranoside (IPTG)-inducible expression vector for the N-terminal His$_6$-tagged cytoplasmic fragment of the wild-type aspartate receptor (TarCF, residues 357–553), which was also used for producing 4Q and 4E mutant TarCF. Plasmids pKJ9[53] (FL), pQM12 (based on pKJ9 with del P1-P2), and pPA770[54] are IPTG-inducible for the expression of CheA and CheW, respectively.

**Protein expression and purification**. E. coli strain RP3098, which lacks all Che proteins and chemoreceptors[55], was transformed with plasmid pKJ9, pQM12, or PPA770 for CheA or CheW expression, respectively. CheA expression was induced at an OD$_{600}$ of 0.6–0.8, with 1 mM IPTG, overnight at 15 °C. CheA was purified using an Affi-gel Blue column (Bio Rad, Hercules, CA) followed by gel filtration on a Superdex 200 column. Further purification with a Mono Q ion exchange column resulted in >99% homogeneity with an overall yield of 50 mg L$^{-1}$ of cells. CheW expression was induced by the addition of IPTG (0.5 mM), at an OD$_{600}$ of 0.4–0.6, at 37 °C. CheW was purified through 20–40% ammonium sulfate precipitation, a DEAE column followed by a MonoQ ion exchange column and a Superdex 75 size exclusion column. His$_6$-tagged wild-type TarCF (His$_6$-TarCF$_{QEQE}$), 4Q TarCF (His$_6$-TarCF$_{4Q}$), and 4E TarCF (His$_6$-TarCF$_{4E}$) were expressed in DH5alpha cells with plasmid pHTCF. TarCF was induced by the addition of IPTG (0.5 mM) at an OD$_{600}$ of 0.4–0.6 at 37 °C and purified with a Ni$^{2+}$-NTA affinity column followed with a mono Q column for quick removal of imidazole. The yield for TarCF was excellent (120 mg L$^{-1}$ of cells).

**Monolayer reconstitution**. A Ni$^{2+}$-NTA lipid-containing monolayer system was used to reconstitute the chemotaxis core-signaling complex arrays, as described previously[16]. A mixture of 9:18:18 µM of TarCF:CheA:CheW in a buffer containing 75 mM Tris-HCl, pH 7.4, 100 mM KCl, and 5 mM MgCl$_2$ was applied to a Teflon well, over which we immediately laid a lipid monolayer containing 2:1 DOPC: DOGS-NTA-Ni$^{2+}$ lipid mixture, at 2 mg mL$^{-1}$ concentration. The monolayer set up was left undisturbed in a humidity chamber overnight. The monolayer specimen was picked up with holey carbon grids, stained with 1% uranyl acetate, and examined with an FEI T12 microscope operated at 120KV.

**CryoET**. Reconstituted monolayers were picked up with perforated R2/2 Quantifoil grids (Quantifoil Micro Tools, Jena, Germany) precoated with 10 nm fiducial gold beads and plunge-frozen using a manual gravity plunger. This method prevents disruption of the monolayer by using single-side blotting that eliminates the contact between the blotting filter paper and the delicate monolayer. The frozen-hydrated electron microscopy grids were loaded into FEI Polara cartridges and imaged under low-dose conditions using a Tecnai Polara microscope (FEI Corp., OR.) operating at 300 kv. A series of low-dose projection images were recorded with tilt angles ranging from 70° to −70° with a Gatan 4 K × 4 K CCD camera

(Gatan, Inc., PA), at a nominal magnification of 39,000×, with a defocus value of 3–7 µm and an accumulated dose of ~60 e$^-$ Å$^{-2}$.

**3D reconstruction, sub-tomogram classification, and averaging**. Collected tilt series were aligned and tomograms were reconstructed using patch tracking and weighted back projection, respectively, in IMOD[56]. To extract sub-tomograms, initial positions and orientations of the receptor complexes were estimated via template matching implemented in emClarity[31]. Both the template and tomograms were low-pass filtered to 4 nm and binned to ~1.2 nm per pixel. Following template matching and sub-volume extraction, the data were randomly split into two groups, which were processed independently for all subsequent steps. Sub-tomogram alignment and classification were carried out using emClarity[31]. Sub-tomograms corresponding to CSUs, as judged by the presence of density corresponding to the CheA dimer and CheW, were retained for the final average.

It should be noted that due to the preferred orientation of monolayer arrays, there exists a resolution anisotropy in the resulting density maps as previously observed. This anisotropy leads to better resolution in the X–Y plane and worse resolution in the Z-direction than the overall resolution indicated. To assess the degree of resolution anisotropy, conical FSCs from the two independent half data sets of CSUs were calculated along each of the principal axes as well as the ten axes bisecting them (Supplementary Fig. 2b)[57]. The averaged density map of CSU was then low-pass filtered according the conical FSCs by using cones with a 42° half-angle, adjusted for any overlapping regions in reciprocal space.

**Component models and homology modeling**. High-resolution structural information for the E. coli chemosensory proteins is sparse. In particular, no high-resolution structures exist of the cytoplasmic portion of Tar. As Tar and Tsr possess highly similar sequences (99% similarity and 86% identity in the modeled region), atomic coordinates from E. coli Tsr (PDB 1QU7)[41] were taken as an initial model for the receptor TOD (residues 340–441). Receptors were truncated near the glycine hinge to reflect the extent of well-resolved receptor density in the 4Q map. Similarly, no high-resolution structures exist for the P3, P4, or P5 domains of E. coli CheA. However, crystal structures of these domains are available from T. maritima CheA, which is sufficiently similar to E. coli to permit the construction of reliable homology models (75% similarity and 42% identity for CheA.P3.P4.P5). Thus, a crystal structure of the T. maritima CheA.P3.P4 dimer (PDB 4XIV)[17] was used to generate atomic coordinates for the corresponding portion of E. coli CheA (residues 264–510), while a crystal structure of the T. maritima CheA.P5-CheW dimer (PDB 4JPB)[42] was used to simultaneously generate coordinates for E. coli CheA.P5 (residues 511–646) and CheW (residues 14–158), thereby, capturing the binding mode at interface 1. All homology models were constructed using Modeller v9.2[58] based on pairwise sequence alignments between the target and template structures. Side chains were subsequently refined using SCWRL4[59] followed by conjugate gradient minimization and all-atom MD.

**MD simulations**. All molecular simulations (i.e., GSA, MDFF, and standard MD) were carried out using NAMD 2.12[60] and the CHARMM36 force field[61]. Equilibrium MD simulations were conducted in the NPT ensemble. Conditions were maintained at 1 atm and 310 K using the Nosé–Hoover Langevin piston (period = 200 fs, relaxation time = 50 fs) and Langevin thermostat (temperature coupling = 5 ps$^{-1}$), respectively. The r-RESPA integrator scheme was employed with an integration time step of 2 fs and SHAKE constraints applied to all hydrogen atoms. Short-range, non-bonded interactions were calculated every 2 fs with a cutoff of 12 Å; long-range electrostatics were evaluated every 6 fs using the particle-mesh-Ewald method.

CSU models were hydrated with TIP3P and neutralized with 150 mM KCl using VMD[62], producing systems containing ~320,000 atoms. Each model was subjected to a conjugant gradient energy minimization (10,000 steps) and a 20-ns NPT equilibration simulation with backbone restraints. Production simulations of CSU models were conducted with weak harmonic restraints (force constant = 0.25 kcal mol$^{-1}$ Å$^{-2}$) placed on the receptor alpha carbons to prevent distortions arising from truncation; the remainder of the complex was not restrained.

**MDFF**. MDFF simulations were performed in the NVT ensemble at 310 K with additional simulation parameters as detailed above. All fittings were carried out in explicit solvent using a scaling factor of 0.1 to couple backbone atoms to the MDFF potential. To prevent the loss of secondary structure as well as the formation of cis-peptide bonds and chirality errors during the fitting process, additional harmonic restraints were applied to the protein backbone using the default force constants provided by the MDFF plugin in VMD. The cascade-MDFF protocol[63] with symmetry restraints[64] was used to fit P4 conformations resulting from clustering analysis of the GSA ensemble; all other fittings utilized the standard, single-density MDFF protocol[65]. Quality-of-fit was assessed in real space by computing the local cross-correlation coefficient (CCC) between the experimental density and a simulated density map derived from the fitted model using the MDFF plugin in VMD[43,66]. To prevent an artificially high assessment of model-map agreement, a threshold was applied to the experimental map to remove relatively low-density values (corresponding to solvent) and only pixels surrounding the localized region of interest in the model (e.g., an individual CheA.P4 domain) were considered in

the calculation[44,66]. UCSF Chimera v1.13[67] was used for map-segmentation and rigid-docking procedures conducted prior to MDFF.

**GSA.** GSA simulations were performed in implicit solvent using previously established parameters for biomolecular applications[47,48,68]. To provide a minimal system that reproduced the structural environment of the P4 domain within the CSU, a CheA.P3.P4.P5 monomer, along with the receptor dimer bound to P5 and the P3 domain from the adjacent monomer, were extracted from the MDFF-refined CSU model (Fig. 3a, Supplementary Fig. 6c). GSA was then used to sample different values of the backbone Phi and Psi dihedral angles of P3–P4 hinge residues 326–328. For this purpose, the GSA approach offers a substantial advantage over standard MD simulation in that much larger conformational transitions may be generated, owing to its Monte Carlo nature, without sacrificing structural fidelity[69]. As the MDFF-refined P4 domain was evidently in a favorable conformation, an initial GSA simulation was conducted to generate several different P4 positions from which to seed an additional round of GSA simulations. The conformations resulting from each of these were combined to produce the analyzed ensemble.

During sampling, the P4 domain and hinge residues (aside from the sampled dihedrals) were held rigid, while the remainder of the structure was held fixed. To prevent unphysical energies arising from steric clashes within the narrow confines of the P3–P4 hinge, which would lead to low acceptance of the conformations proposed by GSA, side chain atoms of residues 326–328 were removed. Additionally, the P4–P5 loop (residues 507–510) was removed to increase sampling efficiency. A minor population of the generated structures had a P4–P5 distance >18 Å (i.e., greater than the length of four residues) and were not included in the analyzed ensemble. Both the side chain atoms and P4–P5 loop residues were subsequently restored using Modeller prior to additional simulation and analysis.

**Data Analysis.** Simulation trajectories were visualized and subjected to basic structural analyses using VMD[62]. Structural clusterings were performed on dissimilarity matrices constructed from pairwise Cα RMSD values using the QCP method implemented in MDAnalysis v0.19.0[49,70]. Clusters were formed via an agglomerative hierarchical approach using the so-called centroid linkage criterion (i.e., UPGMC), employing functions implemented in the python module scipy. cluster.hierarchy. The distance cutoff parameter, which determines the number of returned clusters, was chosen as 20–40% of the maximum distance based on visual inspection of the clustering dendrogram. In cases where it was desirable to select a single cluster representative, the k-medoids algorithm (with k = 1) was used to identify the medoid structure of each identified cluster. Potential salt-bridge interactions were identified by calculating the distance between the outermost side-chain carbon atom of acidic and basic residues using MDAnalysis. If this distance was <7.0 Å for more than a third of the total simulation time, hydrogen bonding between the pair was further analyzed in VMD. Images of molecular structures and density maps used in figures were rendered with VMD.

**Reporting summary.** Further information on research design is available in the Nature Research Reporting Summary linked to this article.

## Data availability

The cryoET density map of the E. coli 4Q CSU is deposited in the EM Data Bank under accession code EMD-10050. The raw tilt series will be deposited EMPIAR database. Coordinates for the corresponding atomic model are deposited in the RCSB Protein Data Bank under accession code 6S1K. All relevant data are available from the authors upon reasonable request.

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

## Acknowledgements

We thank Dr. Robin Corey for critical reading of the manuscript and many valuable suggestions, Professor Mark Sansom for fruitful discussions, and D. Bevan for computer technical support. This work was supported by the National Institutes of Health NIGMS Grants R01GM085043 (P.Z.), RO5GM19559 (J.S.P.) and P41GM104601 (Z.L.-S.), the US National Science Foundation grant PHY1430124 (Z.L.-S.), the UK Wellcome Trust Investigator Award 206422/Z/17/Z (P.Z.), the UK Biotechnology and Biological Sciences Research Council grants BB/S003339/1 (C.K.C., P.J.S., and P.Z.), BB/P01948X/1 and BB/R002517/1 (P.J.S.), the Medical Research Council grant MR/S009213/1 (P.J.S) and the UK Wellcome Trust Technology Award 208361/Z/17/Z (P.J.S.). MD simulations were performed on the Blue Waters supercomputer, which is supported by the National Science Foundation (OCI-0725070 and ACI-1238993) and the state of Illinois. This work is part of the Petascale Computational Resource Grant (ACI-1713784).

## Author contributions

P.Z. conceived the experiments and with C.K.C. designed the experiments. J.M. performed in vitro reconstitution of monolayer arrays of CSUs containing WT, 4Q, and 4E receptor variants. P.Z. and G.Z. collected cryoET data. B.H. and D.S. performed cryoET reconstruction and STA; C.K.C., with P.Z. designed computational approach, constructed structural models, and performed molecular simulations in discussion with Z.L.-S. and P.J.S.; C.K.C. analyzed atomic models and interpreted simulation results in discussion with J.S.P. and P.Z.; C.K.C. and P.Z. wrote the paper with support from other authors.

## Competing interests

The authors declare no competing interests.
