## [Peer Review File · Communications Biology]

Reviewers' comments:

Reviewer #1 (Remarks to the Author):

This manuscript reports an important study that substantially increases our understanding of the structural and dynamical properties of the E. coli chemoreceptor core signaling particle through the combined use of cryo-electron tomography and molecular simulations. The synergistic application of the two approaches follows from a eLife paper from the same group but provides higher resolution insights that comment on how the activity of the CheA histidine kinase may be regulated by receptors. Both the experimental and computational work are impressive in their scope and rigor and the paper is very well written. I do have several suggestions for the authors to consider.:

1. Central to the paper is the identification of several conformational states of the CheA P4 domain. These are generated from a reduced model by generalized simulated annealing and then refined by MDFF against the cryoET density. Although the refinement clusters the P4 conformation in three families, one does wonder how much the cryoET density distinguishes these different states from one another. A cross-correlation metric is referred to on p. 9, but it is not defined. Is this a correlation between the maps and electron density predicted by the structure? If so, it would be useful to know how different this value is for the starting 12 families conformations and the improvement upon refinement.

How is the weight between stereochemical restraints and map agreement determined in the MDFF refinement. This detail should be added to the methods section. At issue is providing the reader with an assessment for how much the cryoET density supports the CheA conformations arising from simulation.

2. Asymmetry in the P3 domain that is linked to P4-P5 interactions is a potentially important insight. How much does P3 asymmetry correlate with the dipped vs undipped conformations of P4? In the conclusion it is suggested that P3 asymmetry causes symmetric changes to the P4-P5 interactions on both subunits:

"Finally, we suggest a novel role for the P3 domain in CheA signaling, namely that through shifts between symmetric and asymmetric positions within the CSU, the P3 domain can alter the interactions and localizations of both P4 domains simultaneously to affect the conformational dynamics of the CheA dimer as a whole."

However, it is also mentioned that the cryoET maps average the subunit density and that P4 motions do not necessarily have to be symmetric in the real particle. Hence, it's unclear why P3 asymmetry would necessarily affect both subunits, it would seem a simpler model to have changes in P3 interactions on one side of the particle affecting P4-P5 contacts for that subunit. Is this also possible?

3. The second section on P.10 describing the beta-sheet to helix transition of the P3-P4 linker could be improved. The wording implies that there is a continuous helix between P3 and P4 in the undipped structure, whereas Fig. 3A seems to show a switch involving one turn of helix at the top of the P4 domain.

"The most striking differences are observed at the P3-P4 interface where disruption of an anti-parallel beta-sheet interaction involving residues M327 and L362 in the "undipped" state allows the formation of a continuous helical connection between the two domains (Fig. 4AB)."

A conserved Pro residue in this linkage would seem to disfavor a continuous helix between the two domains. Also, revise the sentences so that it is unambiguous which state each structural element is in and which interactions need to be broken and then made in the transition.

It is also difficult to comprehend the transition from Fig. 4AB. It may help to color the elements changing conformation differently to distinguish them and also remove the hydrogen atoms, which tend to obscure the side chain conformations.

4. It would be useful to show a superposition of the receptor-to-P5 interaction compared to the receptor-to-CheW interaction. As mentioned in the paper these contacts have different functional consequences, despite being pseudosymmetric. It is quite interesting how they are different.

Minor points:

1. Methods – CheA production – the P3-P4-P5 fragment is not explicitly mentioned, is it purified the same way as full-length? The construct used should be defined.

2. Fig. 4 and Fig. S8 – more typical to depict residue side chains without the hydrogen atoms - removing atom overlaps makes the conformations more straightforward to evaluate.

3. I recommend showing sequence alignments in the supplemental of a representative group of CheA, CheW and receptor sequences (entire sequences are not needed, just key regions) to demonstrate the conservation of the key interacting residues discussed in the text.

4. It is mentioned that some positions in the arrays do not contain CheA density and were thus classified differently in the analysis. What percentage of volumes do not contain CheA density? Do the receptor conformations differ in the absence of the kinase?

Reviewer #2 (Remarks to the Author):

This manuscript describes the use of cryo-electron tomography and sub-tomograph averaging combined with molecular modeling and simulation to investigate the organization and conformational heterogeneity of the core-signaling units that mediate bacterial chemotaxis. The work advances the field. It provides significant new data and new insights that will be of interest to many engaged in the study of molecular mechanisms in sensory signaling.

I have one significant concern plus a few minor comments and suggestions.

Significant concern

1. Stoichiometry of CheW in the core signaling unit. I found the treatment of this stoichiometry somewhat confusing and I expect a knowledgeable reader will also be confused. The stoichiometry of the core unit has generally been identified as one CheA dimer, two receptor trimers of dimers and two CheW's (e.g. ref. 12, 14 and 17). From the initial tomographic characterization of arrays of core signaling units, the existence of CheW-only rings was noted (ref. 17), documented again in ref. 15, and suggested to stabilize the array and thus enhance cooperativity. It could very well be that to assemble stable arrays using truncated chemoreceptors, CheW-only rings are required at essentially all possible positions. However, the extra CheW's do not appear to be necessary for the fundamental activity. Thus it is confusing to indicate that there are four CheW's in the minimal active signaling unit, i.e. the "core" unit. I suggest that the authors' consider this concern and revise the text accordingly.

Other comments.

2. p. 8, line 15. "biochemically" is not really an accurate description of what was demonstrated in live cells. I suggest revision.
3. Some citations appear to be in error, not the most appropriate or could be supplemented. These are as follows:
 - a. p.3, line 17: There are more appropriate references for the difference in kinase activity between the 4E versus 4Q forms of a receptor than ref 7 and 8. For instance, Borkovich et al., 1992; Li and Weis, 2000; Bornhorst and Falke, 2001.
 - b. p.4, line 5: the "nearly three orders of magnitude" statement merits a reference. Ref. 12 documents that value.
 - c. p. 5, line 10: For the reconstitution system, it seems appropriate to acknowledge its initial development, e.g. Shrout et al., 2003; Montefusco et al., 2007.
 - d. p.6, line 18: Add ref 17.
 - e. p. 6, line 19: Ref 31 does not treat ultrastability. I expect the intended reference is Erbse and Falke 2009 Biochemistry.

Reviewer #3 (Remarks to the Author):

This manuscript presents subtomogram averages of in vitro cryoET structures of the 4E, QEQE, and 4Q E. coli core signaling units (CSU). The authors capitalize on the use of elegantly reconstituted receptor arrays on a lipid monolayer, to generate high-resolution tomographic data sets. Due to this and the regularity of the 4Q arrays, they were able to achieve a sub-nanometer reconstruction in this system. Molecular simulations were then used to develop an atomistic model of the entire CSU (Tar, CheA and CheW), based on the density map. Computational approaches were used to assess potential mechanisms of signal transduction through the receptor to its associated cytoplasmic kinase.

From a technical perspective, this study is very impressive, and the results constitute a significant refinement of the model for signal transduction through these molecules. While it would be nice to see predictions of their model tested, I think this opens the way for further, fruitful work in this field.

In the last sentence of the first paragraph of the results/discussion, the authors write:

"This decreasing trend in long-range order is also seen in vivo 7 and is consistent with previous biochemical studies that show the 4E "kinase-OFF" state reduces array cooperativity, likely through disruption of interactions between CSUs 29."

Reference 7 does not support the first part of this statement. In fact, the referenced paper shows the opposite. If this is a mistake then it should be changed. Perhaps I'm mis-understanding what long-range means in this context? Given that this is figure one, I think the authors should make it clear that tomography of in vivo arrays do not appear to "fall apart" in an activity dependent manner. It might be an interesting observation, but I think the interpretation is not on solid ground, given the in vitro nature of this system.

Reviewer #1

“This manuscript reports an important study that substantially increases our understanding of the structural and dynamical properties of the E. coli chemoreceptor core signaling particle... Both the experimental and computational work are impressive in their scope and rigor and the paper is very well written.”

We thank the reviewer for the positive comments on our manuscript.

1. Central to the paper is the identification of several conformational states of the CheA P4 domain. These are generated from a reduced model by generalized simulated annealing and then refined by MDFF against the cryoET density. Although the refinement clusters the P4 conformation in three families, one does wonder how much the cryoET density distinguishes these different states from one another. A cross-correlation metric is referred to on p. 9, but it is not defined. Is this a correlation between the maps and electron density predicted by the structure? If so, it would be useful to know how different this value is for the starting 12 families conformations and the improvement upon refinement.

The cross-correlation metric we have used here and referred to on pg. 9 is one often utilized to assess the quality-of-fit in real-space, namely the Pearson's correlation coefficient. This is computed between the experimental density map and a simulated density map derived from the fitted model. In keeping with convention, we refer to it simply as the “cross-correlation coefficient” (CCC). Full details surrounding computation of the CCC along with an additional reference have now been added to the MDFF section of the Methods (pg 19). In addition, to address the reviewer's questions regarding the initial and final CCC values for the 12 GSA-derived CheA.P4 conformations, we have added a panel to Fig. S7. This new supplementary figure shows the CCC timeseries for each CheA.P4 conformation over the course of their respective refinement simulations. This also shows that the initial P4 conformations display a wide-range of values for the CCC, which are gradually improved by MDFF. Notably, each refinement converges to a CCC value of ~0.4. Therefore, the three classes of P4 conformation identified by clustering the final MDFF frames (i.e., undipped, intermediate, and dipped) fit the 4Q cryoET data equally well and should provide comparable interpretations of the P4 density.

How is the weight between stereochemical restraints and map agreement determined in the MDFF refinement? This detail should be added to the methods section. At issue is providing the reader with an assessment for how much the cryoET density supports the CheA conformations arising from simulation.

Regarding the stereochemical restraints used in our flexible fitting simulations, we applied harmonic restraints of constant strength (i.e., independent of map agreement) in all simulations as per standard MDFF protocol for maps below 5 Å resolution (described in McGreevy et al., 2016). The default force constants provided by the MDFF plugin in VMD were used for the necessary angle, bond, dihedral, and improper potentials. We now include this information in the MDFF section of the Methods (pg 19).

2. Asymmetry in the P3 domain that is linked to P4-P5 interactions is a potentially important insight. How much does P3 asymmetry correlate with the dipped vs undipped conformations of P4? In the conclusion it is suggested that P3 asymmetry causes symmetric changes to the P4-P5 interactions on both subunits:

“Finally, we suggest a novel role for the P3 domain in CheA signaling, namely

that through shifts between symmetric and asymmetric positions within the CSU, the P3 domain can alter the interactions and localizations of both P4 domains simultaneously to affect the conformational dynamics of the CheA dimer as a whole.”

However, it is also mentioned that the cryoET maps average the subunit density and that P4 motions do not necessarily have to be symmetric in the real particle.

Hence, it's unclear why P3 asymmetry would necessarily affect both subunits, it would seem a simpler model to have changes in P3 interactions on one side of the particle affecting P4-P5 contacts for that subunit. Is this also possible?

We thank the reviewer for their questions concerning the asymmetric P3 behavior described in this study. Our simulations demonstrate that P3 adopts asymmetric conformations in both the undipped and dipped CheA conformational classes. Therefore, the ability of the P3 domain to adopt an asymmetric conformation does not appear to be correlated with P4 conformation. To better highlight this aspect, we have altered the discussion on pg 12 and separated the final CheA.P3 positions from “undipped” and “dipped” simulations into two panels in Fig S9.

Regarding the effects of P3 position at the P4-P5 interfaces, we would like to clarify that asymmetries in P3 position cause *differing* P4-P5 interactions within each CheA monomer. In particular, the tight structural coupling between the P3 and P4 domains, both in the “undipped” and “dipped” conformations, causes fluctuations in the position of the P3 four-helix bundle to be transmitted to both P4 domains *simultaneously*. However, these changes are manifested differently at each P4-P5 interface. For instance, if the P3 bundle in an “undipped” CheA drifts towards a particular receptor trimer, it will effectively “push” the P4 domain on the same side and “pull” the P4 domain on the other side, causing an asymmetry between the P4-P5 interfaces. Assuming the P3 domains move as a unit, namely a four-helix bundle, it is difficult to conceive how an asymmetry in the position of the P3 bundle would not affect both subunits. To prevent further confusion, we have altered the discussion on pg 12 to better highlight the strong coupling between the P3 bundle and both P4 domains as well as make clearer the relationship between P3 asymmetry and its effect at the P4-P5 interface.

3. The second section on P.10 describing the beta-sheet to helix transition of the P3-P4 linker could be improved. The wording implies that there is a continuous helix between P3 and P4 in the undipped structure, whereas Fig. 3A seems to show a switch involving one turn of helix at the top of the P4 domain.

“The most striking differences are observed at the P3-P4 interface where disruption of an anti-parallel beta-sheet interaction involving residues M327 and L362 in the “undipped” state allows the formation of a continuous helical connection between the two domains (Fig. 4AB).”

We thank the reviewer for pointing out this ambiguity in our description. We have reworded the cited section to clarify that the “continuous helix” between the P3 and P4 domains is only observed in the “dipped” state. Regarding the structures depicted in Figure 3A, these were derived by generalized simulated annealing simulations, which allow only the backbone dihedrals of the P3-P4 linker to be sampled using Monte Carlo as described in the methods. Therefore, the GSA simulations did not contain a molecular dynamics (MD) component that could permit longer-range changes in secondary structure in response to changes in the relative positions of the domains, and the “dipped” structure depicted in Fig 3A does not show a P3-P4 connecting helix. Rather, the continuous helix was formed during equilibration of the models

using all-atom MD simulations depicted in Fig. 4, and we have updated the opening text of that section to better reflect this.

A conserved Pro residue in this linkage would seem to disfavor a continuous helix between the two domains. Also, revise the sentences so that it is unambiguous which state each structural element is in and which interactions need to be broken and then made in the transition.

As pointed out by the reviewer, a proline (residue 328) would disfavor a continuous helix. Nevertheless, despite this perturbation in the hydrogen-bond network, the connection between the two domains, although slightly kinked, is a continuous helix (Figure R1). We have revised the text to highlight this caveat and better explain the residue-level details of the transition.

Figure R1: Secondary structure timeseries of the P3-P4 connector region (residues 320-332) taken from a representative undipped (left) and dipped (right) CheA monomer. Secondary structure was computed using the STRIDE algorithm; structure codes/colors are as follows: T – turn (teal); E – extended beta strand (yellow); B – beta bridge (gold); H – alpha helix (pink); G – 3_{10} helix (blue); I – π helix (red); C – coil (white).

It is also difficult to comprehend the transition from Fig. 4AB. It may help to color the elements changing conformation differently to distinguish them and also remove the hydrogen atoms, which tend to obscure the side chain conformations.

We have altered Fig. 4AB to better highlight the described transition. As the reviewer has suggested, we have used a separate color for the P3-P4 connector region and removed all non-polar hydrogens from the explicit side chains.

4. It would be useful to show a superposition of the receptor-to-P5 interaction compared to the receptor-to-CheW interaction. As mentioned in the paper these contacts have different functional consequences, despite being pseudosymmetric. It is quite interesting how they are different.

As suggested by the reviewer, we have added a panel to Fig. S10 (formally figure S8) depicting the superposition between the CheW-receptor and CheA.P5-receptor interfaces and highlighting the relative rotation between the two binding modes.

Minor points:

1. *Methods – CheA production – the P3-P4-P5 fragment is not explicitly mentioned, is it purified the same way as full-length? The construct used should be defined.*

We have added information on CheA.P3.P4.P5 in the Methods section.

2. *Fig. 4 and Fig. S8 – more typical to depict residue side chains without the hydrogen atoms - removing atom overlaps makes the conformations more straightforward to evaluate.*

We have altered the relevant panels in Fig. 4 and Fig. S10 (formally Fig. S8) to remove all non-polar hydrogens. Polar hydrogens were kept as they are necessary for showing hydrogen bonding between key residues.

3. *I recommend showing sequence alignments in the supplemental of a representative group of CheA, CheW and receptor sequences (entire sequences are not needed, just key regions) to demonstrate the conservation of the key interacting residues discussed in the text.*

We thank the reviewer for this suggestion. We have generated multi-sequence alignments for CheA, CheW, and Tsr and created WebLogos to better illustrate the sequence conservation within the portions of the proteins contained within our simulations. These graphics have been added to the SI as Figure S8, with residues discussed within the text explicitly labelled.

4. *It is mentioned that some positions in the arrays do not contain CheA density and were thus classified differently in the analysis. What percentage of volumes do not contain CheA density? Do the receptor conformations differ in the absence of the kinase?*

From template matching, the subtomograms were extracted mainly based on the receptor density (predominant signal). These subtomograms contain both class 1 (with CheA density, Figure R2, red oval) and class 2 (without CheA density, Figure R2, purple oval). The ratio of class 1 to class 2 is about 1:2 in an ideal case.

Figure R2: Chemotaxis array of core signaling complex.

Reviewer #2

“The work advances the field. It provides significant new data and new insights that will be of interest to many engaged in the study of molecular mechanisms in sensory signaling.”

We thank the reviewer for the positive review of our manuscript.

1. *Stoichiometry of CheW in the core signaling unit. I found the treatment of this stoichiometry somewhat confusing and I expect a knowledgeable reader will also be confused. The stoichiometry of the core unit has generally been identified as one CheA dimer, two receptor trimers of dimers and two CheW's (e.g. ref. 12, 14 and 17). From the initial tomographic characterization of arrays of core signaling units, the existence of CheW-only rings was noted (ref. 17), documented again in ref. 15, and suggested to stabilize the array and thus enhance cooperativity. It could very well be that to assemble stable arrays using truncated chemoreceptors, CheW-only rings are required at essentially all possible positions. However, the extra CheW's do not appear to be necessary for the fundamental activity. Thus it is confusing to indicate that there are four CheW's in the minimal active signaling unit, i.e. the "core" unit. I suggest that the authors consider this concern and revise the text accordingly.*

We agree with the reviewer that only the two central CheWs (i.e., the one's bound to CheA.P5) appear to be necessary for the fundamental activity of the CSU, while the two flanking CheWs act to "glue" CSUs together into extended clusters through the formation of partial or complete CheW-only rings. Therefore, we now define the fundamental CSU (pg 4) as comprising two CheW monomers and have clarified in the main text (pg 6-7) that our observation of two additional flanking CheW monomers suggests an important functional role for CheW outside the CSU. In addition, we have altered the Figure 1 caption to reflect these changes.

2. *p. 8, line 15. "biochemically" is not really an accurate description of what was demonstrated in live cells. I suggest revision.*

We have removed the word "biochemically".

3. *Some citations appear to be in error, not the most appropriate or could be supplemented. These are as follows:*

a. *p.3, line 17: There are more appropriate references for the difference in kinase activity between the 4E versus 4Q forms of a receptor than ref 7 and 8. For instance, Borkovich et al., 1992; Li and Weis, 2000; Bornhorst and Falke, 2001.*

We appreciate the reviewer for bringing the above articles to our attention and have added the references as suggested.

b. *p.4, line 5: the "nearly three orders of magnitude" statement merits a reference. Ref. 12 documents that value.*

We have added the suggested reference as well as Pan et al., 2017.

c. *p. 5, line 10: For the reconstitution system, it seems appropriate to acknowledge its initial development, e.g. Shrout et al., 2003; Montefusco et al., 2007.*

We appreciate reviewer's suggestion and have added the references.

d. *p.6, line 18: Add ref 17.*

We have added the suggested reference.

e. *p. 6, line 19: Ref 31 does not treat ultrastability. I expect the intended reference is Erbse and Falke 2009 Biochemistry.*

We thank the reviewer for catching this misattributed reference; we have changed it to the correct one as suggested.

Reviewer #3

“From a technical perspective, this study is very impressive, and the results constitute a significant refinement of the model for signal transduction through these molecules.”

We thank the reviewer for the positive review of our manuscript.

In the last sentence of the first paragraph of the results/discussion, the authors write:

“This decreasing trend in long-range order is also seen in vivo 7 and is consistent with previous biochemical studies that show the 4E “kinase-OFF” state reduces array cooperativity, likely through disruption of interactions between CSUs 29.”

Reference 7 does not support the first part of this statement. In fact, the referenced paper shows the opposite. If this is a mistake then it should be changed. Perhaps I’m misunderstanding what long-range means in this context?

We thank the reviewer for raising this important issue. As the reviewer rightly points out, reference 7 (Briegel et al., 2013) does not directly quantify differences in 4Q, WT, and 4E long-range packing order (i.e., the degree of interconnectedness between extended patches of well-packed CSUs). We have, therefore, removed the reference to *in vivo* cryoET work in this context and thank the reviewer for pointing out this mistake. Nevertheless, the authors would like to clarify that we do not feel that reference 7 supports the opposite conclusion either. Rather, the opposite trend to which the reviewer is likely referring is Briegel et al.’s suggestion that the dynamic “order” of the kinase (as measured by the size of the kinase “keel” density) within a single CSU changes with signaling state such that the 4E kinase keel density appears larger (and, thus, more ordered) than the 4Q kinase keel density. To prevent confusion surrounding these different types of order we have specified explicitly in the main text the type of extended inter-CSU-packing to which we refer (pg 5, line 20) and provided additional references (Sourjik and Berg, 2002, PNAS; Li and Weis, 2000, Cell; Frank and Vaknin, 2013, Mol. Microbiol) as further support for the notion that kinase-OFF arrays are generally less cooperative and potentially less ordered.

Given that this is figure one, I think the authors should make it clear that tomography of in vivo arrays do not appear to “fall apart” in an activity dependent manner. It might be an interesting observation, but I think the interpretation is not on solid ground, given the in vitro nature of this system.

The authors would like to clarify that we do not believe the decreased order in the 4E *in vitro* system necessarily implies that monolayer arrays fall apart in an “activity dependent manner.” Rather, there appear to be sizable patches of locally well-packed CSUs in all monolayer systems, and the observed differences in the long-range packing order may reflect functionally-relevant changes in CSU structure (arising from different receptor modification states) that subtly affect their ability to form strong interactions. Nevertheless, the authors completely agree with the reviewer in that we cannot rule out the possibility that the observed change in long-range order is an artifact of the *in vitro* system and, thus, we now state this explicitly in the main text (pg 6, line 4). With respect to *in vivo* cryoET work, as the reviewer highlights, it has

previously been shown (e.g., Briegel et al., 2011, Mol. Microbiol.) that native arrays remain intact and hexagonally-packed during signaling. However, to the authors' knowledge, the detailed long-range packing order of the *in vivo* arrays imaged in these studies has not been examined at the tomogram level and so it is difficult to say whether they exhibit similar variations in the average size of well-packed patches. To highlight this outstanding issue, we have added the above reference and altered the main text (pg 5-6) to reflect that the need for additional study.

REVIEWERS' COMMENTS:

Reviewer #1 (Remarks to the Author):

The authors have addressed my concerns, congratulations on a very fine investigation.

Reviewer #2 (Remarks to the Author):

The revised manuscript has addressed each of the issues raised in my review of the initial version. I have no further concerns.

Reviewer #3 (Remarks to the Author):

I am satisfied with the revisions, and think the text better represent the uncertainty in the field that activity leads to a "falling apart" of receptor arrays. For the sake of clarity, however, I was not referring to changes in the "keel" density observed by Briegel et al., but rather their consistent findings across multiple studies, that whether or not chemoreceptors are activated, they always appear in a tight hexagonal packing, even in the raw data from individual tomograms. It is true, upon further inspection, that Briegel et al, do not explicitly report on the long-range extent of that tight packing.